# The Impact of a Family-Centred Intervention for Parents of Children with Developmental Disabilities: A Model Project in Rural Ireland

**DOI:** 10.3390/children10020175

**Published:** 2023-01-17

**Authors:** Roy McConkey, Pauline O’Hagan, Joanne Corcoran

**Affiliations:** 1Institute of Nursing and Health Research, Ulster University, Belfast BT15 1ED, Northern Ireland, UK; 2Positives Futures, Bangor BT20 5BE, Northern Ireland, UK

**Keywords:** developmental disabilities, family-centred, intervention, parents, social isolation, rural, emotional well-being, practice-based evidence

## Abstract

The greater risk of poor mental health and social isolation, experienced by parents of children with developmental disabilities, is compounded by family circumstances and living in rural settings. Often parents receive little personal support. Family-centred interventions have been recommended internationally for promoting children’s development, as well as boosting parental wellbeing. Yet, in many countries, current service provision is predominately child-focused and clinic-centred. An innovative, family-centred support service was designed and evaluated in a rural county of Ireland. Support staff visit the family home every month for around one year with regular check-ins by phone. The service aims included setting developmental goals for the child that were agreed with parents, alongside actions to address the personal needs of parents and siblings. In addition, community activities are identified or created to promote the social inclusion of the child and family in local communities, as well as locating opportunities for social activities for mothers. To date, 96 families with 110 children have been involved and three monthly reviews have been undertaken of each child’s progress. Baseline measures on parents’ mental health and social isolation were gathered and repeated when parents had completed their involvement with the project, along with qualitative information regarding the parents’ experiences. Most children attained their learning targets, alongside those selected as personal goals by parents; in particular, parents reported their child’s greater involvement in community activities, increased knowledge and skills, and with more confidence and resilience. Significant increases in parental well-being scores were reported, but there was a limited impact on their social participation and that of their child. This evidence-based model of provision is an example of how current social care provision for families who have a child with developmental disabilities could be cost-effectively re-envisioned even in rural areas.

## 1. Introduction

Children with developmental disabilities present many challenges to families. One aspect that is under-recognized in support services made available to such children, is the greater risk of poor mental health and social isolation experienced by the family carers. Their poorer physical and emotional wellbeing is well documented, along with increased stress and intra-family tensions [1,2], all of which can contribute to behavioural issues in managing their child [3]. Parental wellbeing is further compounded by socio-economic circumstances [4], living in rural settings [5] and the recent COVID-19 pandemic [6]. Often, little personal support is available to parents as most educational, health, social care provision is directed at the meeting the child’s assessed needs. Yet, family-centred interventions have been recommended internationally for promoting children’s development, as well as building the resilience and wellbeing of families in meeting the ongoing needs of their child [7].

There is a growing evidence base, internationally, of the value of home-based service delivery and parent-mediated interventions in promoting the development of children with disabilities [8,9]. Moreover, interventions need to commence in the early years of the child’s life [4] and should not be dependent on the child receiving a formal diagnosis, as often happens due to resource constraints [10]. To date, such innovative approaches have attracted more affluent and better educated parents living in urban settings [11].

This paper provides a more full account and an evaluation of an innovative, family-centred support service in a rural county of Ireland provided by an NGO called Positive Futures. An interim report had described the development of the project and the early results of its impact on families [12].

The three main aims of this pioneering support project were:(1)to enhance the children’s social and communication skills and promote their participation in community activities,(2)to provide emotional support to parents and extend their social activities and networks.(3)to boost the resilience and capacity of parents to cope with the challenges they face.

In addition to describing the 96 families and 110 children who participated in the project and the impact it had on children’s development and parental wellbeing, this paper provides an example of how service personnel might undertake an evaluation of family-centred provision, via gathering practice-based evidence. Moreover, the evidence gathered through practice in real-world settings, with all their constraints, will help to confirm the need for new conceptual frameworks to underpin health and social services for children with disabilities.

## 2. Materials and Methods

This section begins with an overview of the service, its delivery and costs. A summary of the various evaluation methods used then follows. For ease of reference, further details of the procedures for gathering information are given when presenting the results obtained in Section 3. Detailed descriptions of the children and families are then presented, including their social and community participation.

### 2.1. Brighter Futures Service

The project focused on families living in county Fermanagh, in the west of Northern Ireland (population 62,000), who had children with a disability diagnosis; however, in later years this was extended to include children who were waiting for assessments, most notably for the Autism spectrum. Locally recruited project staff, trained in family support and child development, visited the family home monthly for around one year with phone calls in between visits. During COVID-19 lockdowns in 2020/21, contact with families was by phone or through Zoom. Learning goals for the children with differing developmental disabilities were agreed with parents, alongside actions to address the personal needs identified for parents and siblings. The social inclusion of the child and family in local communities was encouraged through existing community activities or were provided for them by project staff. A mix of learning activities took place in the family home, alongside outings in the local community for the children to participate in leisure and sport activities. Furthermore, social activities were organized mainly for mothers, but also for siblings and fathers. In addition, opportunities were arranged for families to meet each other socially. The project has been operational for five years and given the resources available, around 20 families have been supported each year; approximately 100 families having been involved over the five years.

#### Costs

New services need to be funded and managed. The UK National Lottery Community Fund covered the total costs of the project for five years from 2017 to 2021, totalling nearly three-quarters of a million pounds sterling (US 940,000); this was developed and delivered by a non-governmental organisation—Positive Futures—which had extensive experience in delivering family support services. The salaries of staff involved were lower than those commonly paid in the UK [13].

### 2.2. Evaluation Methods

The logic model underpinning the family-centred project specified two main outcomes from the theory of change, embodied in the family-centred service, as described above. The first outcome focused on changes in the children and particularly in their social and communication skills; the second focused on improvements in the emotional wellbeing of parent care-givers and greater social connections. The evaluation used a mix of quantitative and qualitative information to assess the extent to which these outcomes had been achieved and also gain insights into the procedures used by the project that had helped to achieve the outcomes.

Further details on the methods used are provided in the results section below; however, in brief, quantitative data came from the records kept by staff on the children and families referred to them. This included details on the intervention targets that were set for the child and for the parents, along with ratings made by the staff and parents on the progress made in achieving the targets after 6 months, 9 months and at around 12 months, when the visits to the family home finished. A variant of goal attainment scaling was used to summarise the progress made within specific developmental domains [14]. Parents also self-completed rating scales on their social participation, and for their emotional and social wellbeing using standardized tools with known reliability and validity [15,16].

Qualitative information was obtained mainly through one-to-one telephone interviews, conducted by an independent researcher (the first author), with parents who had been involved with the project during its third year of operations and with a further sample in its sixth year. The interviews were complemented by self-completed questionnaires provided to all parents and to referrers. However, the qualitative methods and findings are described more fully in an accompanying paper [17].

### 2.3. Description of the Participants

In all, 110 children from 96 families have participated in the project thus far, representing 91% of families referred to the project. Based on the Multiple Indicators of Social Deprivation for N.I. [18], nearly two-thirds of the families (65%) resided in areas that fell within the top 30% of the most deprived areas and only 3% living in the 30% least deprived areas. Moreover, this measure is thought to underestimate the extent of rural deprivation.

For 70 families (73%), both natural parents resided together and a further two were a reconstituted family (2%), while 24 (25%) were single parents. The median number of children in the household was two (range 1 to 7). In all, 31 (32%) families reported having another child with a disability in the family. Of these, 14 families had two or three children who took part in the project. In addition, 14 (15%) families reported that a carer had a disability.

In 84 (88%) families, the mother was reported to be the primary carer of the child with special needs; in seven families (7%), both parents were named, and in five families, (5%), the father was the primary carer.

The mean age of the primary carer was 39 years (range 22 to 61 years). In all, 44 (46%) primary carers had attended higher education; 6 (6%) had left school at 18 years; 30 (31%) had GCSEs and 15 (16%) had left school at 16 years.

Overall, 60 (63%) of the primary carers were not in employment, while 14 (15%) worked full-time, 19 (20%) part-time and two occasionally (2%). However, in 20 (21%) families with two carers, neither were in employment; meanwhile, in 47 (49%) households, both parents were in either fulltime, part-time or occasional employment. Of the 96 families, 46 (48%) were reported to own their own home and 50 (52%) did not.

Families mostly received informal support from other family members, friends and other parents. Overall, the median number of supports that families received was three (range 0 to 9) from a listing provided. However, four families (4%) reported having no informal supports, 15 (16%) had only one form of support, 21 (22 %) had two supports, and 56 (58%) families reported three or more informal supports.

In contrast, only a minority of families received any formal supports, other than schools; these included respite breaks, domiciliary and home visits, and mobility allowances. The majority of families (n = 49: 51%) received none of the formal supports listed, with 31 (32%) receiving one, 12 (12%) receiving two and 4 (4%) receiving three or four of the supports listed.

Parents reported on the various social activities they personally had participated in either during the past month, occasionally, never, or had not wanted to participate in. Comparisons could be made with the leisure activities of a representative sample of adult persons in Northern Ireland that broadly matched the demographics of the project families [19]. Table 1 contrasts the number and percentage of project parents who took part in the activities listed at least once a month, compared to the percentage in the Northern Ireland sample.

Furthermore, 58% of families had not taken a holiday in the past year, compared to 23% of adults in Northern Ireland.

#### Characteristics of the Children

Of the 110 children who completed the project, 78 were boys (71%) and 32 (29%) were girls. Their median age when starting the project was 6.5 years (range 9 months to 13 years).

In all, 62 (56%) were reported to have autism; 33 (30%) had a learning disability and 29 (26%) had other developmental disabilities. In addition, 18 (16%) children had other conditions mentioned. (Note: children could have more than one condition recorded.) A further 18 (16%) children were awaiting a diagnosis. In addition, 31 children had a medical problem; 34 children had visual difficulties (mostly wearing glasses); 17 had physical difficulties and six had hearing difficulties. In all, 44 children (40%) were taking regular medication.

The majority of children had a statement of special educational needs (n = 68: 62%), but this was higher for those with a learning disability (90%), compared to autism (75%) and developmental disabilities (78%).

The children attended the following facilities: 49 (31%) mainstream schools; 41 (37%) special schools or units; 19 (17%) preschools and 7 (6%) were too young to attend schools.

The children’s participation in social activities was recorded from a listing of six available activities, such as youth clubs, sports and after-school clubs. In all, 86 (75%) children did not attend any groups, 17 children (15%) attended one of those listed, and 12 children (101%) attended two or more activities. COVID-19 restrictions may have limited the participation for the cohort who joined the project from March 2020.

## 3. Results—Outcomes for Children and Families

In line with the intended outcomes of the service, as specified in the project’s logic model, we first report on the progress reported by parents and staff on the learning targets that had been set for the children. Second, we examined the personal targets chosen by parents and siblings and the extent to which they had been met. A third focus assessed the changes in parental ratings on their social and emotional wellbeing, given at the start of their involvement with the service and after they had completed their 12 month period of home-based support. This included changes in formal and informal supports, their social engagement and personal wellbeing. A fourth strand to the evaluation, obtained qualitative information about the parental experiences of the service.

### 3.1. Targets Set with Children and Their Progress

Individual learning targets were set for each child according to their development levels. The targets were grouped into the five domains shown in Figure 1. Examples of targets were as follows: ‘Joins in playing Lego with sister’ (Social skills); ‘Uses pictures to communicate choice of drinks’ (Communication); ‘Sleeps in own bed’ (Independence); ‘Accompanies mum to supermarket’ (Confidence); Goes swimming with dad once a week’ (Community); and ‘Dresses self for school each morning’ (Personal care).

The targets were chosen in conjunction with each family involved in the project and these were reviewed at six months, nine months and 12 months (prior to families leaving the project). For each target, a judgement on the child’s progress was made by the parents and project staff using a five-point scale from ‘much better’; ‘better’; ‘the same’; ‘worse’ and ‘much worse’. In order to condense the information relating to the 110 children, the chosen targets were grouped into six domains and within each domain, the number of children whose progress was assessed as ‘much better’ could be ascertained. The number of children varied across domains and once a learning target had been achieved, new targets could be selected for children during the 6–12 month period. Hence, the numbers for whom the targets in each domain were selected may have been made up of different children at each time points. Nonetheless, this approach provides a synthesis of trends across the total sample of children.

Figure 1 gives the percentage of children in each domain for whom the individual target was selected, who were rated to be ‘much better’ at the three review points. These percentages are a conservative estimate as some children were also rated as ‘better’. Three children out of the 110 had been rated as having worsened on one of the targets set at the six month review, but no ratings of worse occurred at the nine and 12 month reviews.

As Figure 1 shows, the improvement in each domain was most marked at the 12 month review, with a steady rise on ‘much better’ ratings during the child’s engagement with the project. Overall, over two-thirds of children were rated as much better in four of the six target areas.

The data also indicates the need for ongoing support for the child’s development beyond the 12 months of the project; as at 12 months, around half of the children had not fully achieved the targets set for them, particularly in respect to personal care and independence.

### 3.2. Targets Set with Parents and Siblings

Similar outcome targets were set for the parents and siblings, as shown in Figure 2. These were made specific depending on the child and family requests. These included, for example, ‘sporting activities for siblings’; ‘provision of picture cards to aid communication’; ‘family days at adventure park’; ‘advice on positive behaviour support’; and ‘mother enrolled on college course’.

For these two groups, the selected targets were rated as ‘fully achieved’, ‘partially achieved’, ‘not achieved’, or a new target was set. Progress was rated by project staff in conjunction with parents. Complete information was available for 102 of 110 children.

Figure 2 summarises the percentage of targets selected for parents and siblings and those which were rated as ‘fully achieved’. However, targets were also rated as ‘partially achieved’, with very few rated as not achieved; thus, the percentages shown in the Figure are a conservative estimate of progress.

The number of targets that were fully achieved rose over the family’s engagement with the project particularly in the final six months, although sibling engagement was evident from the early months of the project.

In sum, the project was perceived by staff and parents as having had a positive impact on the child and on the parents. Nevertheless, certain families would seemingly benefit from ongoing support beyond the 12 months of the project, particularly in building their confidence, resilience and knowledge.

### 3.3. Changes in Parental Ratings on Completion of the Project

After five years of providing the service, a total of 90 (out of 96) carers had completed the monitoring questionnaires prior to starting the project and at the end of their 12 months with the project: a 94% completion rate for the evaluation data.

#### 3.3.1. Changes in Informal and Formal Supports

Across the 96 parents, there had been a slight increase in the mean number of informal supports available to them from family and friends (see Section 2.3), from 3.1 to 3.4, but this was not statistically significant using paired *t*-tests. Likewise there was no difference in the mean number of formal supports the families received, at 0.72. Families who completed the project received marginally more support before and during their time on the project than did families who participated during the COVID-19 restrictions.

There were no significant differences in paired *t*-tests in the low level of formal supports received by the families before and during the project, either pre-COVID-19 or during COVID-19 restrictions.

#### 3.3.2. Changes in Social Engagement

The number of social activities parents engaged in was rechecked at the end of their time with the project (see Section 2.3). For parents who participated in the project pre-COVID-19, there was a small but significant increase in the mean number of activities they participated in monthly (from mean of 1.34 to 2.18: Paired *t*-tests: *p* < 0.05); however, during COVID-19, no difference was reported (mean 1.18 before and 1.04 after).

For the 55 children who took part re-COVID-19, the mean number of social activities they engaged in, either monthly or occasionally, had increased from 4.6 to 5.7, which was statistically significant (paired *t*-tests: *p* < 0.01). However, during the COVID-19 restrictions experienced by 55 children, there was no difference in their social contacts (before 4.2 and after 4.7).

#### 3.3.3. Changes in Parental Wellbeing

The two rating scales, used to assess parental wellbeing, were repeated when families left the project.

On the Edinburgh–Warwick Wellbeing Scale [14], the parents had significantly increased scores at the end of their time with the project (Mean 51.0), compared to the scores at the start (Mean 41.7) (paired *t*-tests *p* < 0.001), and with a large effect size (Cohen’s d = 0.899). At the start, 42 (56% of the 86 parents who completed the scale at the start and end) had below average or very below average scores, but on exiting, 27 had moved to average scores, five to above average scores, while eight remained below average. Overall at the end of the project, 13 parents (15%) had above average scores, 62 parents (72%) had average scores, 10 (12%) had low scores and one (1%) had very low scores.

On the subjective wellbeing measure [15], the total score across the items was calculated (minimum 12 maximum 120). At the start of their involvement with the project, the mean score of 87 parents was 70.1 (range 28–113), and at the end, it was 88.6 (range 55–120). This difference was statistically significant with a large effect size (paired *t*-tests *p* < 0.001: Cohen’s d = 0.985). The increased scores were evident with parents who took part in the project before and during COVID-19 restrictions.

#### 3.3.4. Parental Reactions to the Service

As part of an overall evaluation of the service, at the end of their time on the project, parents anonymously self-completed a brief questionnaire that summarised their experiences with the project. In all, 49 questionnaires were returned (52% response). In addition, 16 parents who had completed the project across different years agreed to be interviewed by the independent evaluator. These interviews focused mostly on their experiences after their involvement ended.

In the questionnaire and interviews, parents were asked to comment on what they felt were the good aspects of the service, how it might have made life better for them as parents, any changes they had seen in the children and in the family, and ideas for how the service might be improved. Additional questions were asked of parents who had left the project up to a year or more ago; these asked how they had managed since, and their perceptions of any lasting impact the project had on the child and family.

Five themes recurred in parental responses, which were identified using thematic content analysis undertaken by the first author and validated with project staff. These are summarised in Figure 3.

The major theme was the family-centredness of the service that, in turn, bolstered the four other themes. Parental confidence in managing their child had improved; the children had developed new skills and both they and the family had become more connected with community activities. Parents valued the information, and the tangible and emotional support given to the family by staff whom had developed trusted relationships with the child and mothers especially. Suggestions for improvements to the project were sparse and tended to focus on the support continuing for more than the 12 months they were usually allocated. More full details of parental perceptions of the project are available in an accompanying paper [17].

## 4. Discussion

The new service was welcomed by families in this rural area, with developmental gains for the children reported and evidence of greater social inclusion found. Likewise, the emotional wellbeing of parents and their confidence was enhanced. In sum, the evaluation confirmed the value of a family-centred response to children experiencing developmental disabilities. Often in disability services, the focus is solely on child outcomes. Yet, an abundance of international research demonstrates that building the competence and resilience of parents, and boosting their personal wellbeing, is crucial to ensuring good outcomes for the children and their physical, social, cognitive and emotional growth [4]. Hence, the project serves as an example to other services of how family wellbeing can be nurtured and its impact assessed. The personal relationships that parents forged with project staff, the home-based support they provided, alongside the parent-focused activities provided by the project, have likely contributed to these outcomes [20,21].

The children attained the developmental targets that were identified through dialogue with parents, children and project staff. Many children had problems with communication and they encountered difficulties in socialising with others. They lacked confidence and were reluctant to become more independent. Through home-based routines and especially through engaging in activities in the community, many children acquired these important life skills, with support from project staff. Such experiences are not easily incorporated into therapy sessions or school classrooms. Hence, more of the latter is not the solution, rather interventions in natural settings will likely be more effective [22].

The project intentionally addressed the extra challenges faced by families living in rural areas [11]. The families came from across the social spectrum, but there was a bias towards more socially disadvantaged families. The parents and the children were often socially isolated, with barely any formal support from services, other than schools. Moreover, a sizeable number of families had little informal support from outside the family. A major contributing factor is the rural setting, particularly when families have no car or mothers cannot drive. The home-based personalised approach, adopted by the project, is essential, given the diversity among the parents and children even within this one small geographical area. Yet, the value of innovative projects, such as Brighter Futures, is not only in the outcomes it provides to families and children. More crucial is the learning that it generates, as to how services can be better shaped to provide cost-effective, emotional and practical supports to families in rural communities who are faced with the challenge of raising a child with developmental disabilities [23].

Although project staff and parents reported high proportions of children attaining the learning targets set for them, some children would benefit from continuing support beyond the 12 month, one-to-one contact that the project provided, as they had not fully achieved their targets. Most parents also reported significant increases in their well-being, but to varying degrees. Hence, project staff need to be sensitive to providing the extra emotional and practical support that some parents may require and over a longer period of time. In sum, there may need to be some flexibility around the time families engage on a one-to-one basis with these projects, which could be achieved through a periodic review of the progress that the project has achieved for the children and for families.

Few changes were apparent in the informal supports available to families or in their social participation. Arguably, one of the legacies of a time-limited engagement with families by professional services is helping them to build ongoing support networks among family and friends. There are some signs that this was starting to happen: parents felt better connected to the local community and knew where to get help and support. That said, there may be cultural as well as practical factors that inhibit parents from seeking support from other parents. Although parents wanted their involvement with the project to continue beyond 12 months, this is ultimately unsustainable, and it prevents other families from enrolling on the project when personnel resources are limited. Hence, other means need to be found for maintaining ongoing support to families, albeit in differing ways. Future research could usefully evaluate services that aim to develop such networks of support [23].

Resource and practical constraints limited the scope of the evaluation. The random allocation of families to take part in the study was not possible, so a selection bias on the part of referrers might be present. In addition, there was no control for changes that may have occurred with families over the passage of time, but the recruitment of a control group of families who had no support would have posed significant ethical challenges. However, the qualitative findings reported by parents, coupled with the changes in the two quantitative indicators, each of which had large effect sizes, are strong evidence that participation in the project significantly impacted the children and families’ well-being.

The value of innovative projects, such as this one, go beyond the outcomes they provide to families and children. In particular, they demonstrate how new models of social support services in rural communities can provide cost-effective, emotional and practical supports to families who are raising a child with developmental disabilities. As others have argued, new conceptual frameworks are needed in childhood disability services that combine the promotion of the child’s development with the needs of parents and the wider family, while taking into account the social and environmental contexts in which they live [7]. Such multi-dimensional approaches will necessitate major transformations to the many current health and social care services for children with developmental disabilities and their families currently provided nationally in the United Kingdom and possibly internationally. Yet, the lower costs of these innovative approaches, allied to the improved outcomes that have been evidenced, suggest that the main challenges lie in moving current systems and traditional staff roles away from outdated modes of thinking and working [24,25].

## 5. Conclusions

Families living in a rural setting welcomed the family-centred service, as evidenced by the high uptake and few drop-outs. The children acquired new skills through the home-based activities and engagement in community activities. In addition, parents reported higher wellbeing scores and some improved social engagements outside of the home, but this did not happen when COVID-19 restrictions were in place. For most families, their 12 month engagement with the project was sufficient, but certain parents would benefit from longer contact. The project is worthy of replication elsewhere and confirms the value of transforming current health and social care for children with disabilities towards more family-centred services.

## Figures and Tables

**Figure 1 children-10-00175-f001:**
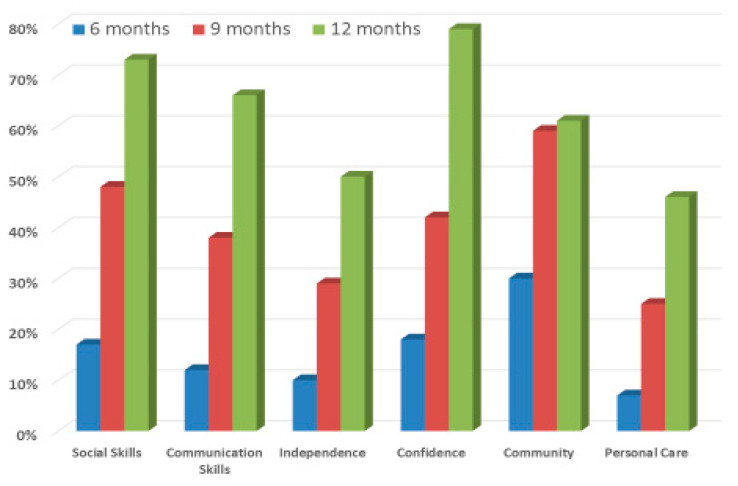
The percentage of children rated as ‘much better’ in each domain at the three reviews.

**Figure 2 children-10-00175-f002:**
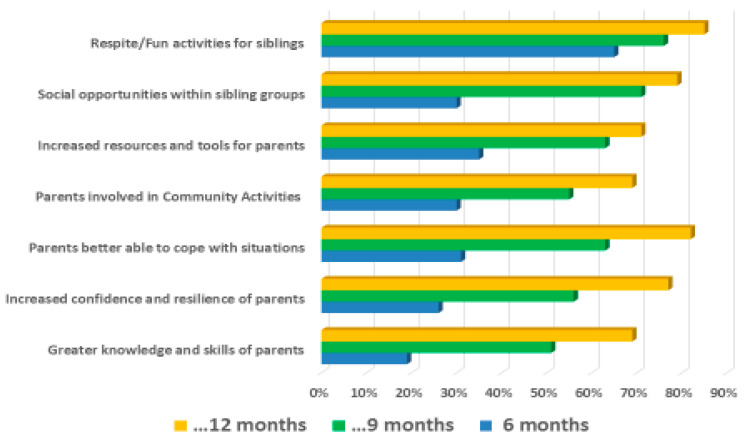
The percentage of parental targets rated as ‘fully achieved’ at the three reviews.

**Figure 3 children-10-00175-f003:**
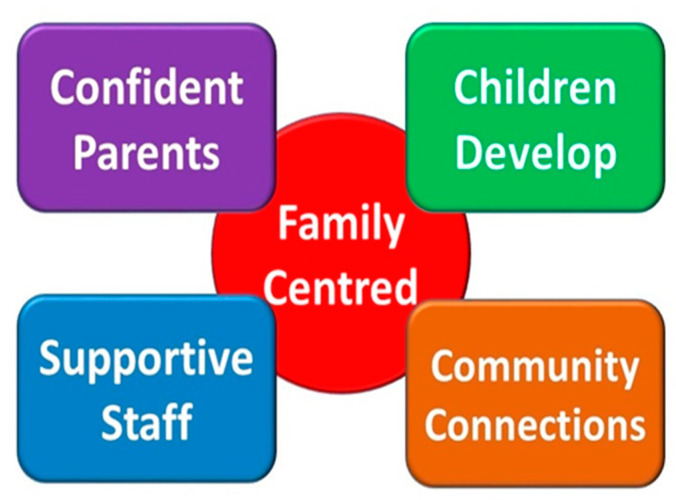
The main themes reported by parents of their experience of the service.

**Table 1 children-10-00175-t001:** The number of parents involved in social activities at least once a month (n = 85) compared to the NI population.

During the Past Year, Have You:	Project Parents	NI%
Had friends/family come to house for coffee/meal etc.	19 (22%)	75%
Been to cinema, theatre, concert	10 (12%)	15%
Attended gym, sports, exercise class	10 (12%)	72%
Been to church/church activities	7 (8%)	30%
Been a volunteer helper	10 (12%)	11%

## Data Availability

The data reported in this paper is available on reasonable requests to the corresponding author.

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
