# Peer review of "The Impact of a Family-Centred Intervention for Parents of Children with Developmental Disabilities: A Model Project in Rural Ireland"

_children, 2023, doi:10.3390/children10020175_

Round 1
Author Response
Please see the attachement

Reviewer 2 Report
The article is more like a repot than a scientific article. It had to be much clearer what is going on in the projects, it is very difficult to understand what is going on in the families. How many people work on this , how long time does each family get visits, what is happening there, what questions are discussed?
Maybe one could make some tables and then discuss since it is so many % and numbers in the text it is difficult to remember when reading,
What is the variation of ages of children, and are there differences in what is made in the family related to ages of children.
The data collected is not always clear, what questions are asked, beside if something has developed to better or worse? Where there any differences in families depending on age of children, number of siblings, families with two parents or only with one? The evidence, more than what parents say themselves is hard to see. And all projects where people are involved and are positive to will lead to the fact that people perceive that something has developed. The evidence have to be clear!
Author Response
The article is more like a repot than a scientific article.
We have made clear it is not a scientific report but an evaluation of a service and description of the form that practice-based evidence could take inline with the special issue in which this paper may appear.
It had to be much clearer what is going on in the projects, it is very difficult to understand what is going on in the families. How many people work on this , how long time does each family get visits, what is happening there, what questions are discussed?
This information is presented in section 2.1. As the reviewer will appreciate, the details s/he requested vary from family-to -family and it would not be possible to describe them for all 96 families in a journal article.
Maybe one could make some tables and then discuss since it is so many % and numbers in the text it is difficult to remember when reading,
We will consider doing this if the editor requests it but our preference is to describe the characteristics of the children and families in the text.
What is the variation of ages of children, and are there differences in what is made in the family related to ages of children.
This information was given in sections 2.2 and 2.3
The data collected is not always clear, what questions are asked, beside if something has developed to better or worse? Where there any differences in families depending on age of children, number of siblings, families with two parents or only with one?
Our goal was to evaluate the overall impact of the service and to do so we had to find ways of grouping the information gained from individual children and families who had received that were specific to their needs. It would not then be possible or indeed appropriate to examine the outcomes measures in terms of the predictor variables named by the reviewer.
The evidence, more than what parents say themselves is hard to see. And all projects where people are involved and are positive to will lead to the fact that people perceive that something has developed. The evidence have to be clear!
The reviewer does not specify what evidence s/he felt needed to be clearer. In much social and health service evaluations the major outcome is participant self-report. We augmented this by staff engagement in ratings and use of multiple indicators. Like most services, funds to undertake more thorough evaluation of impact are not available. The other two reviewers appreciated the value of gathering practice-based evidence that we had describded.
Reviewer 3 Report
Dear authors,
the paper is interesting but it is very complicated, not sound in their methodological parts. The reader can not come with a clear picture of your intervention program and the results. Moreover, the instruments and baseline/final measures are not explained or presented (even in an Appendix). I make some suggestions for improvement.

Round 2
Reviewer 1 Report
See attachment.

Author Response
I have replied to the set second set of comments by this reviewer.

Reviewer 3 Report
Dear authors, the 2n version of the paper is really improved but there are still thinks to do. I have added detailed comments on the paper to help your work. I hope they are sufficient.

Author Response
I had replied previously to these comments.